# Osmotic Stress Responses, Cell Wall Integrity, and Conidiation Are Regulated by a Histidine Kinase Sensor in *Trichoderma atroviride*

**DOI:** 10.3390/jof9090939

**Published:** 2023-09-16

**Authors:** Gabriela Calcáneo-Hernández, Fidel Landeros-Jaime, José Antonio Cervantes-Chávez, Artemio Mendoza-Mendoza, Edgardo Ulises Esquivel-Naranjo

**Affiliations:** 1Unit for Basic and Applied Microbiology, Faculty of Natural Sciences, Autonomous University of Queretaro, Queretaro 76230, Mexico; calcaneo@ifc.unam.mx (G.C.-H.); landeros@uaq.mx (F.L.-J.); jose.antonio.cervantes@uaq.mx (J.A.C.-C.); 2Departamento de Genética Molecular, Instituto de Fisiología Celular, Universidad Nacional Autónoma de México, Ciudad de México 04510, Mexico; 3Faculty of Agriculture and Life Sciences, Lincoln University, Lincoln 7647, New Zealand; artemio.mendoza@lincoln.ac.nz

**Keywords:** sporulation, cell wall integrity, histidine kinase, stress cellular, MAPK signaling

## Abstract

*Trichoderma atroviride* responds to various environmental stressors through the mitogen-activated protein kinase (MAPK) Tmk3 and MAPK-kinase Pbs2 signaling pathways. In fungi, orthologues to Tmk3 are regulated by a histidine kinase (HK) sensor. However, the role of *T. atroviride* HKs remains unknown. In this regard, the function of the *T. atroviride* HK Nik1 was analyzed in response to stressors regulated by Tmk3. The growth of the Δ*nik1* mutant strains was compromised under hyperosmotic stress; mycelia were less resistant to lysing enzymes than the WT strain, while conidia of Δ*nik1* were more sensitive to Congo red; however, ∆*pbs2* and ∆*tmk3* strains showed a more drastic defect in cell wall stability. Light-regulated *blu1* and *grg2* gene expression was induced upon an osmotic shock through Pbs2-Tmk3 but was independent of Nik1. The encoding chitin synthases *chs1* and *chs2* genes were downregulated after an osmotic shock in the WT, but *chs1* and *chs3* expression were enhanced in ∆*nik1*, ∆*pbs2*, and ∆*tmk3*. The vegetative growth and conidiation by light decreased in ∆*nik1*, although Nik1 was unrequired to activate the light-responsive genes by Tmk3. Altogether, Nik1 regulates responses related to the Pbs2-Tmk3 pathway and suggests the participation of additional HKs to respond to stress.

## 1. Introduction

*Trichoderma* species are common inhabitants of the soil and rhizosphere, which sense harmful environmental changes such as osmotic stress, temperature fluctuations, pH changes, nutrient limitations, oxidative stress, light, and wound [1,2]. These stressors act as cues to initiate protective responses driving *Trichoderma* to conidiation, and the activation of mitogen-activated protein kinase (MAPK) signaling cascades has been related to *Trichoderma’s* morphological and physiological changes [3].

In yeast, the MAPK Hog1 participates in osmotolerance [4], oxidative stress [5], heat shock resistance [6], cell wall integrity [7], and cadmium tolerance [8], and it is conserved across fungal and animal species [9]. In *Trichoderma*, the ortholog to yeast Hog1, called Tmk3/ThHog1, plays a role in conidiation, resistance to high osmotic pressure, cell wall integrity maintenance, heavy metal toxicity, and oxidative stress [10,11,12]. In *T. harzianum*, ThHog1 is phosphorylated and localized in nuclei under osmotic stress, and it is required for antagonistic activity against plant pathogens [12]. In *T. reesei*, Tmk3 is involved in cellulase production and regulates the synthesis of chitin and β-1,3-glucan [11], while in *T. atroviride*, Tmk3 and the MAPKK (mitogen-activated protein kinase kinase) Pbs2 regulate cellular stresses, conidiation, and the expression of genes regulated by light [10]. Although Pbs2 has been proposed as the direct activator of Tmk3 in *T. atroviride*, upstream components of the Pbs2–Tmk3 pathway have not been characterized.

The upstream elements of the yeast Hog1 signaling pathway include the MAPKK Pbs2, which activates the MAPK Hog1 by phosphorylation of Thr^174^ and Tyr^176^ residues [4] and three MAPKKKs: Ste11, Ssk2, and Ssk22 [13]. The osmosensors Sho1, Msb2, Hkr1, and Opy2 activate the Ste11-Pbs2-Hog1 branch through interaction with the MAPKKKK Ste20 [14]. At the same time, the redundant MAPKKK Ssk2 and Ssk22 are activated via the Sln1-Ypd1-Ssk1 two-component signal transduction system (TCS) [13].

In filamentous fungi, Hog1 orthologs are specific targets of the TCS. In fungi, TCS consists of a histidine kinase (HK) sensor, a histidine-containing phosphotransferase (HPt), and a response regulator (RR) protein [9,15,16]. In the genome of *T. atroviride*, twelve HKs were identified, and one of them was assigned as class III [17], a homolog to Nik-1/Os-1, which contains HAMP domain repeats in its amino terminus and is conserved in filamentous fungi and pathogenic yeasts [18]. Nik-1 was identified as a member of the osmotic response signal transduction cascade in *Neurospora crassa* and was designated Os-1 [19]. In *N. crassa*, HK Os-1/Nik-1 is essential for hyphal development, cell wall integrity, conidiation, and fludioxonil sensitivity [20,21]. Although Os-1 is considered a possible element in the osmotic stress response pathway, it is dispensable for the phosphorylation of the Os-2 MAPK in response to osmotic and heat shock stress [21]. In *Aspergillus fumigatus*, HK NikA contributes to osmotic adaptation, conidiation, hyphal morphology, cell wall structure, and fungicide stress responses, but it appears to have roles that are independent of the MAPK SakA [22].

To identify players upstream of the Pbs2-Tmk3 signaling pathway, the main aim was to analyze the role of the HK Nik1 and its potential relationship with the MAPKK Pbs2 and MAPK Tmk3 of *T. atroviride*. We found that Nik1 regulates osmotic stress responses, cell wall integrity, and asexual reproduction. Altogether, our results suggest that the activation of the MAPK Tmk3 may require additional HKs in response to external stimuli.

## 2. Materials and Methods

### 2.1. Strains and Culture Conditions

*T. atroviride* IMI 206040 was used as the wild type strain (WT). ∆*pbs2*-7 and ∆*tmk3*-13 mutants were previously reported [1,10]. All strains were propagated on a potato dextrose agar (PDA; DIFCO) at 27 °C in light or dark conditions. PDA plates supplemented with 100 μg mL^−1^ Hygromycin B (Invitrogen, Carlsbad, CA, USA) were used as a selection medium or supplemented with different chemicals for cellular stress analysis.

### 2.2. Construction of ∆nik1 Mutants

The *nik1* sequence of *T. atroviride* (EHK40885.1) was used to design primers (Table 1) to replace the open reading frame (ORF) with the hygromycin phosphotransferase gene *(hph*) selectable marker. Following the double-joint PCR methodology [23], the *nik1* 5′ and 3′ flanking regions were amplified using primers P*nik1*-F–PQ*nik1*-R and TQ*nik1*-F–T*nik1*-R, respectively. The *hph* gene was amplified using primers Hyg-F–Hyg-R from plasmid pCB1004 as a template [24]. The three PCR fragments were joined in a second PCR reaction through chimeric sequences (primers PQ*nik1*-R and TQ*nik1*-F). The product of the second PCR was used as a template to amplify the selection cassette using the nested primers N5*nik1*-F–N3*nik1*-R, and the product was directly used for protoplast transformation. Protoplast isolation and transformation were carried out as described before [25]. The PCR reactions were run using Platinum™ Taq DNA Polymerase High Fidelity (Invitrogen, Carlsbad, CA, USA) with the following conditions: first step at 94 °C/2 min, 35 cycles at 94 °C/15 s, 60 °C/15 s, 68 °C/1 min per kb, and a final extension at 68 °C/5 min. All *T. atroviride nik1* transformants were subjected to six rounds of single spore isolation. The gene replacement by double homologous recombination was verified by PCR and RT-PCR. Genomic DNA of the WT and mutant strains were prepared according to Raeder and Broda [26]. Complementary DNA (cDNA) was synthesized as described before [25]. The DreamTaq DNA Polymerase (Thermo Scientific, Waltham, MA, USA) was used for PCR reactions, and the program was as follows: first step at 95 °C/3 min, 35 cycles at 95 °C/30 s, 60 °C/30 s, 72 °C/1 min per kb, and a final extension at 72 °C/5 min.

### 2.3. Cellular Stress Assays

A conidial suspension (1 × 10^5^ conidia/mL) of the WT, ∆*nik1*, ∆*pbs2*, and ∆*tmk3* strains was prepared to evaluate stress tolerance in conidia. Then, 5 µL of the conidial suspension was inoculated on PDA plates supplemented with 0.5% Triton X-100, supplemented with or without the different stressors: sorbitol or NaCl to test osmotolerance; CdCl_2_ for cadmium toxicity; Congo red to challenge cell wall integrity; and H_2_O_2_ or menadione for oxidative stress tolerance. Then, the cultures were incubated at 27 °C for six days.

To evaluate stress tolerance in mycelia, precultures were generated on PDA plates with 2 µL of a conidial suspension (1 × 10^8^ conidia mL^−1^) of the WT and mutant strains and incubated for 48 h at 27 °C in darkness. Mycelial plugs from the precultures (0.5 cm) were inoculated on PDA supplemented with or without the stressors indicated above for conidia. The plates were incubated at 27 °C for four days. All stress assays were performed in triplicate.

### 2.4. Sensitivity to the Cell Wall Lytic Enzyme Assay

To analyze the impact of cell wall integrity in the WT, ∆*nik1*, ∆*pbs2*, and ∆*tmk3* strains, a conidial suspension (1 × 10^6^ conidia mL^−1^) of the respective strains was inoculated in 100 mL of a GYEC liquid medium (1.5% glucose, 0.3% yeast extract, 0.5% casein, and pH adjusted to 5.5 with KOH) in a 500 mL Erlenmeyer flask. The culture was incubated in a constant orbital agitation (160 rpm) at 27 °C for 18 h. The mycelium for each strain was filtered, and 0.1 g of the mycelium was transferred into a 50 mL conical tube containing 7 mL of osmotic solution (50 mM CaCl2, 0.5 M mannitol, 50 mM MES, pH 5.5) and 6 mg mL^−1^ of lysing enzymes from *T. harzianum* (Sigma-Aldrich, St. Louis, MO, USA). The mycelium was then incubated in orbital agitation (120 rpm) at room temperature for 2 h to form protoplasts. Protoplasts were filtered through sterile Miracloth, washed with 2 mL of osmotic solution, and collected by centrifugation (8000× *g* rpm). Then, the supernatant was discarded, the protoplast pellet was resuspended in an osmotic solution, and protoplasts were counted in a Neubauer chamber. The assay was carried out in triplicate.

### 2.5. Gene Expression Induced by Osmotic Stress

A conidial suspension (1 × 10^6^ conidia mL^−1^) of the WT and mutant strains was inoculated in 45 mL of PDB medium (potato dextrose broth) in a 250 mL Erlenmeyer flask covered with aluminum foil and incubated in constant orbital agitation (160 rpm) at 27 °C for 48 h. To test the effect of osmotic stress in gene expression of chitin synthase encoding genes (*chs1*-*chs8*), the β-1,3-glucan synthase (*fks1*) and blue light-induced genes *blu1*, *grg2*, and *env1*, 5 mL of 5 M NaCl were added to a liquid medium under red safelight, which were incubated for an additional 5, 15, 30 or 60 min. Then, mycelia were filtered, frozen in liquid nitrogen, and macerated. Mycelia grown in PDB without stress were used as a control. Total RNA was extracted with TRIzol^®^ Reagent (Invitrogen, Carlsbad, CA, USA), following the specifications for RNA isolation. The cDNA was synthesized using 1 µg total RNA using RevertAid Reverse Transcriptase (Thermo Scientific, Waltham, MA, USA). PCR reactions were carried out to analyze the gene expression using the primers listed in Table 2, and they followed the next PCR program: first step at 95 °C/3 min, 35 cycles at 95 °C/30 s, 60 °C/30 s, 72 °C/1 min per kb, and a final extension at 72 °C/5 min.

Primers were designed to produce amplicons around 300 bp. For gene expression, cycle numbers listed in Table 2 were determined experimentally by choosing a cycle where the PCR product exponentially increased before reaching the maximum signal for each transcript. For this approach, we made an end-point PCR that increased for five cycles for each reaction, from 20 to 45 cycles (20, 25, 30, 35, 40, and 45). Then, the signals were analyzed by electrophoresis to determine the cycle when the signal was saturated. PCR was repeated, decreasing one cycle in each reaction from the sutured signal cycle; e.g., if the signal was saturated at 35 cycles, PCR was repeated from 30 to 35 (30, 31, 32, 33, 34, and 35 cycles). Once we determined the exponentially growing cycles, we chose the middle point or two cycles before saturation started. It was essential to make this approach using the treatment with the highest transcript levels to detect differences between the two treatments better. This analysis was carried out with transcripts extracted 15 min after an osmotic shock or 30 min after a blue light pulse, as previously described [10], to easily detect differences among transcript levels.

### 2.6. Radial Growth Measurement

Mycelial plugs (0.5 cm) from precultures were transferred under a red safelight onto PDA plates. Three plates were incubated under constant white light (0.586 µmol m^−2^ s^−1^), whereas three additional plates were kept in darkness. The plates were incubated at 27 °C for 48 h, and a photograph registered the growth. The radial growth was measured using ImageJ software version 1.52a (https://imagej.net/ij/index.html, accessed on 15 September 2023). The assay was carried out in triplicate.

### 2.7. Conidial Production Induced by Light

Mycelial plugs from precultures (0.5 cm) were inoculated on PDA plates in triplicate and incubated at 27 °C for seven days under constant white light conditions (0.586 µmol m^−2^ s^−1^). To analyze conidiation induced by a blue light pulse, colonies grown at 27 °C for 36 h were exposed to a blue light pulse (152.4 μmol m^−2^) and incubated for 48 h in darkness. Then, conidia were harvested with 16 mL of sterilized water and counted in a Neubauer chamber. The assay was performed in triplicate.

### 2.8. Expression Assays of Light-Regulated Genes

The WT and mutant strains were photoinduced, as described before [25]. Total RNA extraction and cDNA synthesis were performed, as described, for expression analysis of cell wall-related genes. The primers used are listed in Table 2.

### 2.9. Statistical Analysis

The graphs and statistical tests were made with GraphPad Prism version 5 (GraphPad Software, San Diego, CA, USA). The graph shows the average of the experiments plus the standard deviation. Data were analyzed by ANOVA followed by the Tukey–Kramer post-test.

## 3. Results

### 3.1. Identification and Deletion of the T. atroviride nik1 Gene

The *T. atroviride* Nik1 has an identity of 63.5% to the HK Os-1 of *N. crassa* (AAB01979.1), 61.1% to the NikA of *Aspergillus nidulans* (EAA60822.1), 63.5% to the HIK1 of *Pyricularia grisea* (BAB40947.1), 62.2% to the Bos1 of *Botrytis cinerea* (ATZ45935.1), and 57.7% to the Nik1 of *Candida albicans* (AOW30628.1). The *nik1* gene encodes a protein of 1324 aa (Figure 1A), containing six repeats of HAMP domains in its amino terminus (191–246, 275–327, 367–419, 459–511, 551–603, 643–695 aa) and the typical domains encoded by fungal HK genes: a phosphoacceptor (710–775 aa) with a conserved His^720^ residue (Figure 1B), an ATP-binding domain (822–942 aa), and a RR receiver domain (1089–1205 aa) with a conserved Asp^1139^ residue (Figure 1C). These data suggest that *T. atroviride* Nik1 has all the conserved motifs for a functional class III HK.

To investigate the role of HK Nik1 in *T. atroviride* IMI206040, the *nik1* gene was replaced by the *hph* selectable marker (Appendix A). Six stable mutants were obtained after six rounds of single-spore isolation on PDA supplemented with 0.5% Triton X-100 and 100 μg mL^−1^ Hygromycin B. The *nik1* gene replacement was confirmed by PCR, which detected the integration of the drug-resistance marker gene at the corresponding locus and the absence of the *nik1* ORF. P*nik1*-F–Hyg-R primers were used to amplify a 2.7 kb fragment comprising the *nik1* 5′ region and the *hph* gene only in the ∆*nik1* strains (Appendix A). A 2.8 kb fragment comprising the *hph* gene and *nik1* 3′ region was amplified in ∆*nik1* strains using Hyg-F–T*nik1*-R primers (Appendix A). To corroborate the lack of the *nik1* coding region, 1*nik1*-F–3*nik1*-R primers only amplified a 0.6 kb fragment of the *nik1* ORF in the WT strain (Appendix A). All results confirmed that the *nik1* ORF was successfully knocked out in the six strains analyzed.

### 3.2. Nik1 Is Involved in Osmotic Stress Resistance

In ascomycetes, Nik1 orthologs generally participate in hyperosmotic stress responses [18]. To examine the role of the *T. atroviride* Nik1 in tolerance against high osmotic pressure, the conidia and mycelia of the WT, ∆*nik1*, ∆*pbs2*, and ∆*tmk3* strains were subjected to high concentrations of NaCl and sorbitol. In the conidia stage, 100 mM NaCl inhibited the growth of ∆*nik1*, ∆*pbs2*, and ∆*tmk3* strains, and 200 mM sorbitol reduced the growth of ∆*nik1* strains. At the same time, that concentration was lethal for ∆*pbs2* and ∆*tmk3* strains (Figure 2A). In mycelia, 200 mM NaCl and 400 mM sorbitol slightly impaired the radial growth of ∆*nik1*, and NaCl was more noxious than sorbitol. In contrast, the ∆*pbs2* and ∆*tmk3* strains did not grow at those osmolyte concentrations (Figure 2B). These results suggest that Nik1 has a role in hyperosmotic stress tolerance in *T. atroviride*, and because the ∆*nik1* strains were not as hypersensitive as ∆*psb*2 and ∆*tmk*3 strains, these findings suggest that additional HKs may be participating in osmotic stress responses along with Nik1.

### 3.3. Nik1 Regulates the Cell Wall Integrity Maintenance of T. atroviride

An alteration in the architecture and elasticity of fungal cell walls provokes sensitivity to osmotic stress [28]. To assess if Nik1 has a role in cell wall integrity, Congo red was used as a cell wall disruptor. Compared to the WT, ∆nik1 strains could not grow at 75 µM of Congo red. In contrast, ∆*pbs2* and ∆*tmk3* strains were unable to grow at lower Congo red concentrations (Figure 3A). However, when using mycelia plugs, ∆*nik1* mycelia displayed a phenotype similar to the WT strain (Appendix A). In contrast, as reported previously, mycelial growth of ∆*psb2* and ∆*tmk3* strains was compromised in Congo red media under constant illumination [10]. To validate if ∆*nik1* strains have a defect in cell wall integrity, we tested the sensitivity to cell wall lysing enzymes in the WT and mutant strains (Figure 3B). The deletion of the *nik1* gene increased the sensitivity to cell wall lysing enzymes in contrast to the WT, producing a double number of protoplasts, while the lack of *pbs2* and *tmk3* genes affected cell wall composition, producing three times more protoplasts than the WT. These results suggest that Nik1 is involved in cell wall integrity maintenance via the MAPK Tmk3 signaling pathway.

Furthermore, we assessed if additional stressors regulated by Tmk3 are affected in *Δnik1* strains. Thus, tolerance assays in oxidative stress, cadmium stress, heat shock, and UV light irradiation were performed. The ∆*nik1* strains showed no sensitivity in response to these stressors (Appendix A).

### 3.4. Osmotic Stress Provokes Changes in Gene Expression through the Tmk3 MAPK Pathway

In *T. harzianum*, two homologs to Grg-1 of *N. crassa*, encoding a glucose-repressible protein, were induced in response to high osmotic stress [12]. In contrast, *T. atroviride* demonstrated that the *blu1* and *grg2* genes, homologs to *N. crassa* Grg-1, are induced through the Pbs2–Tmk3 pathway after blue light stimulus [10,29]. To determine if the gene expression of *blu1* and *grg2* is induced after high osmotic stress in *T. atroviride*, the mycelia of the WT strain were grown in PDB under dark conditions and subjected to high osmotic stress. After 500 mM NaCl treatment, the *blu1* and *grg2* genes reached their maximum transcript levels at 15 min. Then, the *blu1* expression decreased, and the *grg*2 transcript levels remained high after 60 min of treatment (Figure 4A,B). The *env1* gene, which encodes a PAS/LOV domain protein, is activated by blue light stimulus independent of Tmk3 [10,30]. Transcripts of the *env1* gene were not detected during the experiment, indicating that the *blu1* and *grg2* genes were stimulated explicitly in response to an osmotic shock in *T. atroviride*.

Considering that a cell wall protects from a hyper-osmotic shock in fungi [31], *T. atroviride* genes encoding chitin synthases (*chs1-8*) and β-1,3-glucan synthase (*fks1*) were identified (Table 2), and their expressions were evaluated in response to high osmotic stress. Transcript levels of *chs1*, *chs2*, *chs3*, and *chs6* genes decreased after 15 min of treatment with 500 mM NaCl (Figure 4C), while *chs4*, *chs5*, *chs7*, and *fks1* expression remained without apparent change in the time analyzed. *chs8* gene expression was not detected under our experimental conditions.

In order to test if the Tmk3 pathway regulates the expression of *blu1*, *grg2*, and cell wall-related genes via Nik1, the WT and ∆*nik1*, ∆*pbs2*, and ∆*tmk3* strains were subjected to high osmotic stress. After 15 min of treatment, the *blu1* and *grg2* genes were induced in response to osmotic stress only in the WT and ∆*nik1*, but not induced in the ∆*pbs2* and ∆*tmk3* strains, indicating that the Tmk3–Pbs2 pathway regulates their expression independent of Nik1 (Figure 5A). Furthermore, *env1* transcripts were undetected, indicating that activation was specifically stimulated by osmotic shock. Further, the expression of *nik1* and *tmk3* was undetected in ∆*nik1* and ∆*tmk3* strains, respectively, corroborating that the mutant strains lack the corresponding genes. Consistently, transcript levels of *chs1* and *chs2* genes decreased in the WT, whereas *chs1* and *chs3* were higher in ∆*nik1*, ∆*pbs2*, and ∆*tmk3* strains in comparison to the WT strain (Figure 5B). However, their expression was still responsive to a hyperosmotic shock. These data suggest that the Nik1 could regulate *chs1* and *chs3* genes through the Tmk3 MAPK pathway by acting as a repressor.

### 3.5. The HK Nik1 Is Involved in Mycelial Growth and Asexual Reproduction

In *T. atroviride*, the MAPK Tmk3 regulates mycelial growth and conidiation triggered by light. The vegetative growth of the ∆*nik1* strains was analyzed (Appendix A), and the radial growth was reduced by 10% in darkness and 20% under constant white light in contrast to the WT, whereas the growth of the ∆*pbs2* and ∆*tmk3* strains was reduced by 30% and 40%, respectively, compared to the WT strain (Appendix A). Furthermore, when compared to the WT, the conidial production stimulated by a blue light pulse in the ∆*nik1* strains was 90% less. In contrast, in constant white light, the ∆*nik1* strains produced 50% less conidia than the WT (Figure 6A). The Δ*tmk3* and Δ*pbs2* strains produced 80% less conidia in constant white light in comparison to the WT strain, as reported previously [10]. These results suggest that Nik1 is required for vegetative growth and conidiation induced by light in *T. atroviride*, possibly by a mechanism regulated through the Tmk3 pathway. To evaluate if Nik1 is a specific regulator of the photoconidiation process, the conidial production of the ∆*nik1* strains in response to the wound was analyzed, and it was observed that conidiation was also reduced in ∆*nik1*, similar to ∆*pbs2* and ∆*tmk3* strains (Appendix A), suggesting that the Tmk3 pathway plays a general role in *T. atroviride* for asexual reproduction. Based on these results, we examined the expression of light-responsive genes regulated by the Tmk3 MAPK pathway after 30 min of a blue light pulse (Figure 6B). Nik1 is dispensable for *blu1*, *grg2*, and *env1* expression regulated by light. Taken together, our results indicate that Nik1 regulates conidiation, but the activation of light-responsive genes through the Pbs2–Tmk3 pathway occurs by a mechanism independent of Nik1.

## 4. Discussion

Prokaryotic organisms use the TCS to sense and respond to a wide range of environmental signals, and the signal transduction is mediated by the phosphorylation of histidine and aspartic acid residues [18]. In eukaryotic cells, an environmental cue is generally transduced by modifications of serine/threonine and tyrosine residues of signaling proteins. However, plants and fungi possess genes that code for proteins of the TCS. One of the most characterized HK in fungi is Nik-1/Os-1, which was first reported in *N. crassa* and acts upstream of the MAPK Os-2 pathway and regulates osmotic stress responses [21]. In this research, the role of the HK Nik1 in *T. atroviride* was analyzed in response to diverse cellular stressors regulated by the MAPK Tmk3. Our results suggest that Nik1 participates as a receptor in response to osmotic stress, and another HKs could cooperate as osmosensors to transduce the signal through Tmk3. The first HK identified in fungi was Sln1 in yeast [32], which is an osmotic stress sensor that possesses transmembrane domains and is categorized as class VI and is conserved among filamentous fungi [18]. However, a sensor homolog to yeast Sln1 in *T. reesei* is nonessential to cope with osmostress [33]. It would be interesting to explore the role of Sln1 (EHK50841.1) in *T. atrorivide* to elucidate if it is also a player in response to osmotic stress along with Nik1 or another HK that has gained this new role.

A highly saline environment not only imposes osmotic pressure on cells but also intracellular ion toxicity [34,35]. From this perspective, filamentous fungi have a more complex osmoregulatory system than yeast and are capable of discerning stress caused by high levels of salt or sugar. We observed that in *T. atroviride*, higher levels of NaCl were more toxic than sorbitol, suggesting that HK Nik1 is also required to survive intracellular toxicity by NaCl. On the contrary, a *Pyricularia oryzae* strain lacking the *HIK*1 gene, which codes for an HK class III, is more sensitive to high levels of sorbitol than NaCl [36], suggesting that *P. oryzae* can also distinguish if the osmostress is caused by high concentrations of sugar or salt through Hik1. In *Magnaporthe oryzae*, the ∆*Mosln1* strain is more susceptible to salt stress, whereas ∆*Mohik1* is affected by high doses of sorbitol. Interestingly, a phosphorylation signal of MoHog1p was detected in a ∆*Mohik1*/∆*sln1* strain after an osmotic shock, suggesting that more osmosensors participate in the MoHog1p MAPK cascade [37]. A *Beauveria bassiana* strain lacking a class VIII HK was affected by high NaCl and sorbitol concentrations, and a movement of HK8:GFP to the inside of hyphal cells from the cell periphery after 30 min exposure to NaCl was observed [38]. The class VIII HKs have in common a GAF and a phytochrome domain [18]. According to the reports mentioned above, exploring the role of *T. atroviride* Sln1 and Phy1 (EHK44075.1) in osmoadaptation would be interesting.

The mechanisms for osmotic stress tolerance in fungi are based on producing osmolytes in the cytoplasm to balance osmotic pressures and cell wall reinforcement [39]. Our results on Congo red and lytic enzyme resistance assays suggested that Nik1 maintains the cell wall integrity in *T. atroviride*, most likely with other signaling pathways, including other HKs, as observed in other fungal systems. In response to other stressors, such as oxidative, cadmium, thermal shock, and UV irradiation, Nik1 is dispensable. Other fungal HKs are involved in cell wall formation. The PmHHK1 class X of *Penicillium marneffei* regulates sporulation, polarized growth, and cell wall composition [40]. *C. albicans chk1*∆ strains are sensitive to high doses of Congo red and Calcofluor white [41]. *M. oryzae*, ∆*Mosln1*, ∆*Mohik5* (class V), and ∆*Mohik9* (class XI) strains released more protoplasts than the wild type strain after an application of lysing enzymes, but the ∆*Mohik6* (class X) strain was more resistant of the effect of lysing enzymes than the parental strain [37]. In *B. bassiana*, the growth of a ∆*hk5* strain, which lacks a class V HK, was affected by Congo red and a strain lacking the homolog to Nik1 [38]; these data suggest a complex role among HKs in fungi. Our results open the possibility that the Sln1 and the HK class V (EHK42346.1), X (EHK40776.1), and XI (EHK40433.1) cooperatively could participate in the cell wall homeostasis of *T. atroviride*.

Chitin is an important structural element of the cell wall in fungi, and chitin synthases are essential enzymes for their construction [42]. Strains lacking *chs* genes encoding chitin synthases were analyzed in *T. atroviride*, and the growth and conidiation of the Δ*chs1*, Δ*chs2*, Δ*chs5*, and Δ*chs7* strains were severally compromised [43]. Consistently, the transcript levels of *chs1* and *chs2* decreased after 15 min of an osmotic shock, and the *fks1* encoding β-1,3-glucan synthase was constitutively expressed, suggesting that chitin is the leading targeted polysaccharide to modulate the cell wall architecture in responses to osmostress. In *Ustilago maydis*, β-1,3-glucan synthase was expressed constitutively, while the expression of four of the eight genes that code for chitin synthases decreased after treatment with 1 M NaCl [44]. In addition, the expression of five *chs* genes from *Penicillium digitatum* evaluated under osmotic stress conditions was downregulated [45]. The studies mentioned above suggest that in response to osmotic shock, *chs* genes have differences in transcription levels, as suggested *by* our results.

Furthermore, the *C. albicans* Hog MAPK pathway mediates the regulation of the *CHS* gene expression. The chitin synthase activity in a ∆*hog*1 strain is elevated compared to WT cells [46]. Accordingly, we analyzed the expression of *chs1*, *chs2*, *chs3*, and *chs6* genes in the ∆*nik1*, ∆*pbs2*, and ∆*tmk3* strains of *T. atroviride*. Our results revealed that *chs1* and *chs3* showed higher transcriptional levels in the mutant strains in an osmostress-independent manner, and after a hyperosmotic shock, *chs2* and *chs3* genes remained upregulated. The results suggest that Nik1, Pbs2, and Tmk3 act as negative regulators of these genes to maintain homeostasis in cell wall synthesis; consequently, the absence of *nik1*, *pbs2*, and *tmk3* genes affects the cell wall integrity and generates cells more sensitive to osmotic stress.

The *T. atroviride blu1* and *grg2* genes, which encode glucose-repressible proteins, are induced in response to an osmotic shock and blue light pulse through the MAPK Tmk3 and MAPKK Pbs2 pathways. However, HK Nik1 is dispensable for their expression in response to both stressors. MAPK Tmk3 is phosphorylated in response to blue light; once activated, it regulates the expression of *blu1* and *grg2* genes [10]. Yeast Hog1 is also activated by phosphorylation and is translocated into the nucleus to regulate gene expression, although the nuclear import of Hog1 is not essential for osmoadaptation [47]. The results suggest that Nik1 uses two different mechanisms; it acts as a negative modulator of gene expression and as an osmostress-response activator; this is probably carried out by nuclear and cytosolic functions reported for Hog1. In addition, our results suggest that other signaling pathways, including HKs, collaborate with Nik1 in response to environmental stresses, revealing a functional redundancy. However, Nik1 is required for blue light conidiation but not gene expression induced by blue light. Consistent with this proposal, in *A. nidulans*, it was demonstrated that an HK, FphA, is stimulated by red and blue light to activate SakA, a homolog to Tmk3 [48]. Also, in *M. oryzae*, the HPt gene *YPD1* was expressed in response to light [49], suggesting that alternative histidine kinases could regulate light responses and support the possible participation of additional proteins upstream of Tmk3 for asexual reproduction.

In conclusion, our results suggest that the *T. atroviride* Nik1 regulates tolerance to osmotic stress, cell wall integrity, and asexual reproduction. The phenotypes and gene expression indicate a collaborative regulation among histidine kinases to regulate the Tmk3 signaling pathway. Finally, our results suggest that Nik1 modulates stress-independent gene expression through the Tmk3 MAPK signaling pathway.

## Figures and Tables

**Figure 1 jof-09-00939-f001:**
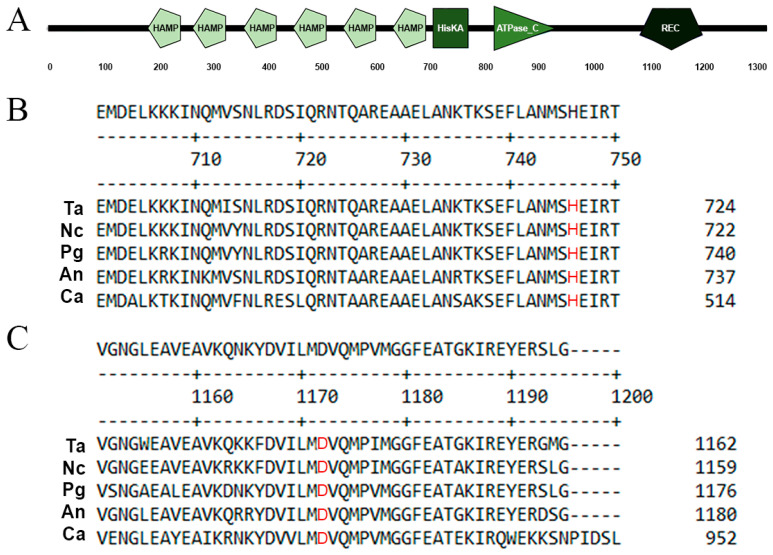
Nik1 amino acid sequence analysis. (**A**) Domains of the *T. atroviride* Nik1 protein predicted by SMART [27]. HAMP (histidine kinases, adenylyl cyclases, methyl-accepting chemotaxis proteins, and phosphatases) domain; HisKA, phosphoacceptor domain; HATPase_c, ATP-binding domain; REC, receiver domain. (**B**,**C**) Amino acid sequence alignment to identify the conserved phosphorylable His^720^ residue in the HisKA motif (**B**) and the Asp^1139^ residue in the REC domain (**C**). Alignment was carried out using MegAlign software version 7.1.0 (DNAStar) by the Clustal W Method. Ta, *T. atroviride* IMI206040; Nc, *N. crassa*; Pg, *P. grisea*; An, *A. nidulans*; Ca, *C. albicans*.

**Figure 2 jof-09-00939-f002:**
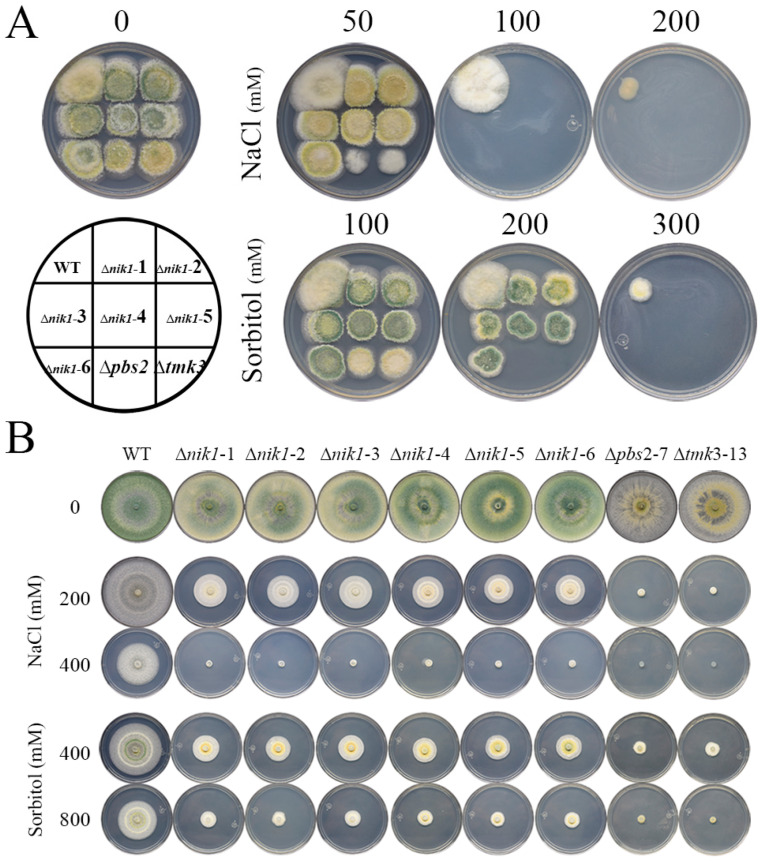
The HK Nik1 regulates tolerance to high osmolarity challenges. (**A**) Tolerance to osmotic stress in the conidia of WT, ∆*nik1*, ∆*pbs2*, and ∆*tmk3* strains. Drops of 500 conidia of the WT and mutant strains were inoculated on PDA plates plus different concentrations of the indicated stressors. Plates were incubated at 27 °C for four days. (**B**) Tolerance to osmotic stress in the mycelia of WT, ∆*nik1*, ∆*pbs2*, and ∆*tmk3* strains. NaCl and sorbitol were added to PDA media at the concentrations indicated. Strains were incubated at 27 °C for four days. All experiments were carried out in triplicate.

**Figure 3 jof-09-00939-f003:**
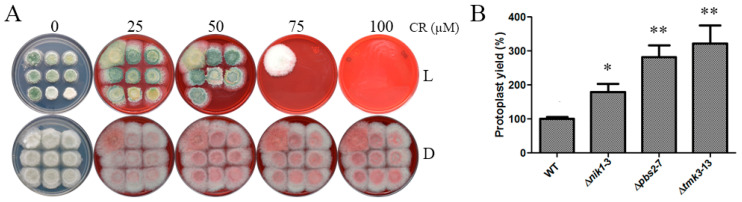
The HK Nik1 plays a role in cell wall integrity. (**A**) Tolerance to Congo red in the conidia of WT, ∆*nik1*, ∆*pbs2*, and ∆*tmk3* strains. Drops of 500 conidia of the WT and mutant strains were inoculated on PDA plates with Congo red at the indicated concentrations, incubated at 27 °C for four days in constant white light and darkness, and pictures were taken. The strain order is indicated in Figure 2. (**B**) Sensitivity to cell wall lysing enzymes of WT, ∆*nik1*, ∆*pbs2*, and ∆*tmk3* strains. The total production of protoplasts was determined using a Neubauer chamber. The mean value is represented in bars ± SEM of three independent experiments analyzed with the Tukey–Kramer method (α = 0.001). * and ** indicate mean values that are statistically different from the control.

**Figure 4 jof-09-00939-f004:**
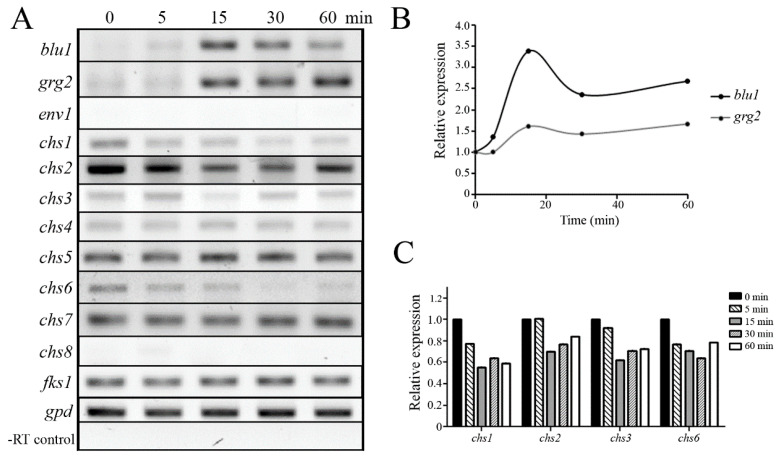
Expression analysis in response to osmotic stress. (**A**). The mycelia of the WT strain were challenged by adding 0.5 M NaCl. Then, samples were collected at the indicated time. Total RNA was extracted from samples and used to synthesize cDNA. As a loading control, the *gpd* gene was amplified. In the same reaction for cDNA synthesis, the RNA of samples without transcriptase was used as a template to amplify *gpd* as a negative control. (**B**,**C**) Semiquantitative transcript levels were determined according to signal intensity from two biological replicates and plotted. The expression level for each gene was normalized by dividing the control *gpd* signal. Relative expression was adjusted to the unit in the control (0 min) without an osmotic shock and compared with the treatment at different time points.

**Figure 5 jof-09-00939-f005:**
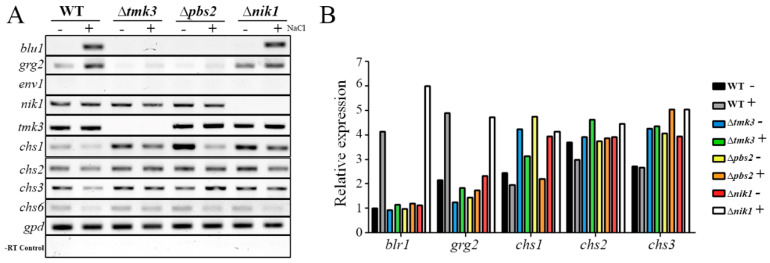
Role of Nik1, Pbs2, and Tmk3 on gene expression regulated by osmotic stress. (**A**) Total RNA was extracted from the mycelia of WT and mutant strains that were non-stressed (−) or 15 min after an osmotic shock by 0.5 M NaCl (+) and used to synthesize cDNA. In the same reaction for cDNA synthesis, (1) RNA of samples without transcriptase was used as a negative control and (2) *gpd* was used as a loading control (cDNA template). (**B**) Signal intensity was quantified from two biological replicates and plotted as indicated in Figure 4. The expression level for each gene was normalized and divided by the control *gpd* signal. Relative expression was adjusted to the unit in the WT without osmotic stress (−).

**Figure 6 jof-09-00939-f006:**
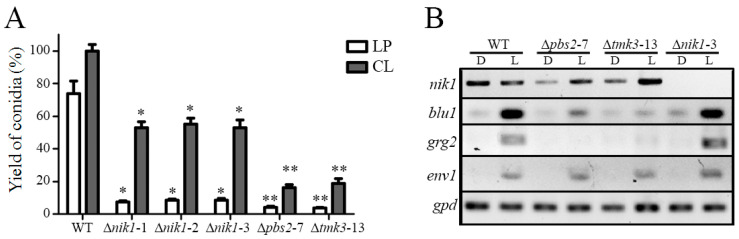
Asexual reproduction and gene expression regulated by light in ∆*nik1* mutants. (**A**) Conidial production of the WT strain and the ∆*nik1*, ∆*pbs2*, and ∆*tmk3* mutant strains. Mycelial plugs of the indicated strains were inoculated on PDA plates, incubated at 27 °C in darkness for 36 h, and then exposed to a blue light pulse (152.4 μmol m^−2^) (LP), or grown for 7 days in constant white light (0.586 μmol m^−2^ s^−1^) (CL). The mean value of conidia yield is represented in bars ± SEM of three independent experiments analyzed with the Tukey–Kramer method (α = 0.05). * and ** indicate mean values that are statistically different from the control. (**B**) Expression analysis by RT-PCR of light-induced genes in the WT, ∆*pbs2*, ∆*tmk3*, and ∆*nik1* strains. D, darkness; L, light.

**Table 1 jof-09-00939-t001:** Primers used for gene deletion.

Gene	Primer	Sequence (5′ → 3′)	Target Region
*nik1* (EHK40885.1)	P*nik1*-F	CTT GCA GGC ACA TCC TTG ACG	5′ flanking region
	PQ*nik1*-R	TGC TCC TTC AAT ATC AGT TAA CGT CGA TCA CGC TCG GCT CGG GTA AGC	
	TQ*nik1*-F	CCC AGC ACT CGT CCG AGG GCA AAG GAA TAG ACA GAA CCA GCT CAT CCA GAC C	3′ flanking region
	T*nik1*-R	CTC TTA TCC ACC TTC CGT CCG	
	N5*nik1*-F	TCG CCT GAG ACT TCC AAG ACG	Nested *nik1* primers
	N3*nik1*-R	TCA AGC CTG CAG CTC TCT CTC	
*hph* (AEJ60084.1)	Hyg-F	GAT CGA CGT TAA CTG ATA TTG AAG GAG CA	*hph* marker
	Hyg-R	CTA TTC CTT TGC CCT CGG ACG AGT GCT GGG	

**Table 2 jof-09-00939-t002:** Primers used for the identification of mutants and analysis of gene expression.

Gene	Primer	Sequence (5′ → 3′)	Number of PCR Cycles
*nik1* (EHK40885.1)	1*nik1*-F	CGA ATG TCG AGG GCA AGT GG	35
	3*nik1*-R	GTG AAG TCG CCG TCT GTA GC	
*tmk3* (EHK43400.1)	*tmk3*-F	GTT TGG TCT TGT CTG CTC TGC G	35
	*tmk3*-R	GCA GGT CGG TTC CGA GAA GC	
*gpd * (EHK49005.1)	*gpd*-F	GCC GAT GGT GAG CTC AAG GG	26
	*gpd*-R	GGT CGA GGA CAC GGC GGG A	
*blu1* (EHK44319.1)	q*blu1*-F	CGT TGG CTC TCG CCT GAC C	27
	*blu1*-R	GAA CGC CAT TGA AGG CCT CG	
*grg2* (EHK50625.1)	*grg*2-F	GAT TCC ATC AAG CAG GGT GCC	27
	*grg2*-R	GTT TAG ATA GCC TGC TTG TGG G	
*env1* (EHK44161.1)	*env1*-F	GCC AAA ATG GTT CCT TCA GGG TC	27
	*env1*-R	GTT TGG TCG AGA CAC AAG TCG G	
*chs1* (EHK39721.1)	*chs1*-F	CTG ACG TTC CCG ACA CTG TTC C	40
	*chs1*-R	TGC CAG TCC ACC AGC GAC G	
*chs2* (EHK46279.1)	*chs2*-F	GAT TCG CGC CAA CCA TGT CG	32
	*chs3*-R	GTA GGA TAA AGC ATC AAC CGA GG	
*chs3* (EHK39554.1)	*chs3*-F	CCT CAG GCA GTA GCT ACC AC	40
	*chs3*-R	CGT GGA CAG TGG AGG CAG G	
*chs4* (EHK40657.1)	*chs4*-F	GTC CGC GAT CTC TGT GGC AC	32
	*chs4*-R	ACC AAG AGT GTG CGG TGA CG	
*chs5* (EHK48324.1)	*chs5*-F	CCG TGC ATG GCT AAG ACT TGG	35
	*chs5*-R	GAG TCG GGT GTG TAG ATG CAG	
*chs6* (EHK48325.1)	*chs6*-F	GTT CGA CTG GGT CAG AAT GGC	40
	*chs6*-R	CCA TTG GAG AAC TGA GAC GAC G	
*chs7* (EHK48360.1)	*chs7*-F	TCA TCA CAG CCG CAC CAG C	35
	*chs7*-R	GAG TCG ATT GAT GCA GAG AAC C	
*chs8* (EHK41601.1)	*chs8*-F	TCT TCG GAA ATG TCT CGC ACC	40
	*chs8*-R	CGG AGC CTT GCC TCT TCC	
*fks1* (EHK39881.1)	*fks1*-F	CTC TTC TGG TTA TTG CCC AGT C	35
	*fks1*-R	GTT GCT TGG TTG TAA CAG TCG G	

## Data Availability

Not applicable.

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
