# Peer review of "Osmotic Stress Responses, Cell Wall Integrity, and Conidiation Are Regulated by a Histidine Kinase Sensor in Trichoderma atroviride"

_jof, 2023, doi:10.3390/jof9090939_

Round 1
Reviewer 1 Report
Calcáneo-Hernández et al. investigated the role of the HK Nik1 and its potential relationship with the MAPKK Pbs2 and MAPK Tmk3 of T. atroviride. They proved Nik1 regulates osmotic stress responses, cell wall integrity, and asexual reproduction. They showed that Nik1 regulates responses related to the Pbs2-Tmk3 pathway and suggests the participation of additional HKs to respond to stress. The work is interesting and provides useful information. In my opinion, the design of experiments, scientific writing, and presenting data is appropriate.
Author Response
Reviewer 1.
Calcáneo-Hernández et al. investigated the role of the HK Nik1 and its potential relationship with the MAPKK Pbs2 and MAPK Tmk3 of T. atroviride. They proved Nik1 regulates osmotic stress responses, cell wall integrity, and asexual reproduction. They showed that Nik1 regulates responses related to the Pbs2-Tmk3 pathway and suggests the participation of additional HKs to respond to stress.
The work is interesting and provides useful information. In my opinion, the design of experiments, scientific writing, and presenting data is appropriate.
Response: Thanks so much for your time in reviewing our manuscript.
Reviewer 2 Report
The manuscript “Osmotic stress responses, cell wall integrity, and conidiation are regulated by a sensor histidine kinase in Trichoderma atroviride” describes new insights into the function of the sensor histidine kinase Nik1 in Trichoderma atroviride. The results are relevant to understanding the MAPK signaling pathway in T. atroviride and its physiological response to changes in the external microenvironment. The technical shortcomings of the work are listed below, and the authors should fix them.
General
- Change ml to mL in all the article text.
-Change µl to µL in all the article text.
- Put a space between the number and the degree symbol in all the article (°C).
- Symbols for genes are italicized. Please correct this in all the text.
Materials and Methods
-In section 2.3, lines 105-106: the authors say, “Then, 5 μl of the conidial suspension was inoculated on PDA plates supplemented with 0.5% Triton X-100”. The addition of Triton X-100 to the culture medium can retard the growth of the fungus. Therefore, wouldn't the addition of this reagent be one more variable in addition to the different stressors that affect growth?
-In section 2.3, line 112-113: the authors say “Mycelial plugs from the precultures (0.5 cm) were inoculated on PDA supplemented”.
Does transferring 0.5 cm of the pre-culture guarantee that the same amount of mycelium from the different strains is being transferred to the plate with PDA supplemented or not with stressors? This is fundamental to comparing the results between the mutants and the wild-type.
- In section 2.5, line 135: write the name of the blue-light-induced genes in this sentence.
- In section 2.5, table 2: the authors say “For gene expression, cycle numbers were determined experimentally, choosing a cycle where the PCR product exponentially increased before reaching the maximum signal for each transcript. This analysis was carried out with transcripts extracted 15 min after an osmotic shock or 30 min after a blue light pulse to easily detect differences among transcript levels”. Delete this sentence in the legend of table 2 and rewrite it in the methodology of section 2.5. Describe in more detail how this analysis was performed, cite an article that has performed the same type of analysis.
- In section 2.5, table 2: How was the maximum signal of each transcript determined? How many cycles were tested for each gene?
Results
- In section 3.1, Figure 1: Chance A, B and C to a, b and c in the legend.
- In section 3.1, line 196: change ml-1 to mL-1.
- In section 3.2: Plot a graph showing the measured radius of colony growth for all strains under all tested conditions. Apply statistical analysis and show if it is statistically significant.
- In section 3.2, lines 210-211: Calculate and write the percentage of the reduction compared to the parental.
- In section 3.2, figure 2: Identify in Figure 2a which colonies are the mutants, which mutants they are, and which colony is the wild-type.
- In section 3.2, figure 2: Change A and B to a and b in the legend.
- In section 3.3: repeat the analysis of conidia growth in congo red. The different strains tested grew so much in four days that it is difficult to analyze the effects on the growth of the strains. It would be interesting to cultivate them in less time.
-In section 3.3, lines 231-232: Calculate and write the percentage of the reduction compared to the parental.
-In section 3.3, figure 3: Improve the quality of the image. Change A and B to a and b in the legend. Write in the legend what are both D and L in the image.
- In section 3.3, figure 3, line 244: the authors say “The strain order is as indicated in Fig. 2”. Delete this sentence in the legend of figure 3 and write the strain order in the Figure 3.
-In section 3.3, figure 3, line 248: the authors say “An asterisk indicates mean values that are statistically different from the control” but in figure 3b the statistic is represented with different letters.
-In section 3.3, lines 250-252: Plot a graph showing the measured radius of colony growth for all strains under all tested conditions. Apply statistical analysis and show if it is statistically significant.
-In section 3.4, lines 254-255: rewrite this sentence.
-In section 3.4 line 279. Write that this was the negative control.
-In section 3.4, figure 4: Specify what figures B and C are in the legend because these two graphs are different. In addition, statistically analyze these two graphs, showing the standard deviation of each point and showing whether the expression is statistically different at each point.
- In section 3.4, lines 289-294: I think this statement could be phrased more concisely without repeating "Again".
- In section 3.4, line 301: Write that this was the negative control.
- In section 3.4, figure 5b: statistical analysis needs to be included, otherwise, authors cannot claim whether there is any significant difference between samples.
- In section 3.5, lines 323-325: Can the authors revise the result narrative and discussion in this regard?
Discussion:
-In this section, the authors could also describe the roles of histidine kinase SNL1 already characterized in Trichoderma reesei.
-lines 344-363: Can the authors revise the result narrative and discussion in this regard?
-lines 377-379: Can the authors revise the result narrative and discussion in this regard?
-line 381: change chk1Δ to Δchk1.
- lines 397-402: Can the authors revise the result narrative and discussion in this regard?
-line 421: perhaps it is more appropriate to change the word prove to suggest.
Supplementary material
Figure S2 - Plot a graph showing the measured radius of colony growth for all strains under all tested conditions. Apply statistical analysis and show if it is statistically significant.
Figure S2b- Identify in Figure which colonies are the mutants, which mutants they are, and which colony is the wild-type.
Figure S3 - Plot a graph showing the measured radius of colony growth for all strains. Apply statistical analysis and show if it is statistically significant. Identify in Figure which colonies are the mutants, which mutants they are, and which colony is the wild-type.
no comments
Author Response
Reviewer 2:
The manuscript "Osmotic stress responses, cell wall integrity, and conidiation are regulated by a sensor histidine kinase in Trichoderma atroviride" describes new insights into the function of the sensor histidine kinase Nik1 in Trichoderma atroviride. The results are relevant to understanding the MAPK signaling pathway in T. atroviride and its physiological response to changes in the external microenvironment. The technical shortcomings of the work are listed below, and the authors should fix them.
We thank you for taking the time to review our manuscript. Your comments were so well appreciated.
General
- Change ml to mL in all the article text.
Response: We changed ml to mL along the manuscript.
-Change µl to µL in all the article text.
Response: Changed µl to µL along the manuscript.
- Put a space between the number and the degree symbol in all the article (°C).
Response: The space a space between numbers and °C along the manuscript.
- Symbols for genes are italicised. Please correct this in all the text.
Response: Thanks so much for your observation. In the new version of the manuscript, the gene names are italicised.
Materials and Methods
-In section 2.3, lines 105-106: the authors say, "Then, 5 μl of the conidial suspension was inoculated on PDA plates supplemented with 0.5% Triton X-100". The addition of Triton X-100 to the culture medium can retard the growth of the fungus. Therefore, wouldn't the addition of this reagent be one more variable in addition to the different stressors that affect growth?
Response: You are right; the detergent might cause stress based on the formation of compacted colonies. However, we used the detergent after verifying the absence of any effect in the nik1 mutants compared to the wild type, as is observed in Figure 2A in zero salt concentration, where no affection in growth is observed. Thus, this result allowed us to use Triton X-100 to compare several genotypes and their phenotypes in the same plate, a method commonly used by us and others (https://doi.org/10.1111/mmi.13355; https//doi.org/10.1007/s42770-020-00329-7; https:/fungalbiolbiotech.biomedcentral.com/articles/10.1186/s40694-019-0078-5). In conclusion, our data shown in Figure 2A indicates that Triton X-100 did not have interference in the growth of both nik1 and WT strains, and results were also similar to other deletions from the same pathways (e.g., tmk3 and pbs2) Tmk3.
-In section 2.3, line 112-113: the authors say "Mycelial plugs from the precultures (0.5 cm) were inoculated on PDA supplemented".
Does transferring 0.5 cm of the pre-culture guarantee that the same amount of mycelium from the different strains is being transferred to the plate with PDA supplemented or not with stressors? This is fundamental to comparing the results between the mutants and the wild-type.
Response: Thank you for the reviewer's suggestion. We used plugs of the same size cut from the active growth area (the youngest zone at the border of the colonies). This approach has been used in several publications, and we have always seen a very low variation in triplicate, indicating that inoculum has uniformly been used (https://doi.org/10.1016/j.crmicr.2022.100139, https://doi.org/10.1111/mmi.13355 and https//doi.org/10.1007/s42770-020-00329-7).
- In section 2.5, line 135: write the name of the blue-light-induced genes in this sentence.
Response: Thank you, gene names were indicated.
- In section 2.5, table 2: the authors say "For gene expression, cycle numbers were determined experimentally, choosing a cycle where the PCR product exponentially increased before reaching the maximum signal for each transcript. This analysis was carried out with transcripts extracted 15 min after an osmotic shock or 30 min after a blue light pulse to easily detect differences among transcript levels". Delete this sentence in the legend of table 2 and rewrite it in the methodology of section 2.5. Describe in more detail how this analysis was performed, cite an article that has performed the same type of analysis.
Response: Thank you for the suggestion; the explanation was moved to section 2.5 and cited a paper where this approach was performed.
- In section 2.5, table 2: How was the maximum signal of each transcript determined? How many cycles were tested for each gene?
Response: For this approach, first, we made an end-point PCR increasing five cycles for each reaction, from 20 to 45 cycles (20, 25, 30, 35, 40, and 45). Then, the signals were analysed by electrophoresis to determine the cycle when the signal was saturated. The PCR was repeated, decreasing one cycle in each reaction from the sutured signal cycle, ej, if the signal was saturated at 35 cycles, PCR was repeated from 30 to 35 (30, 31, 32, 33, 34, and 35 cycles). Once we determined the exponentially growing cycles, we chose the middle point or two cycles before saturation started. It was essential to make this approach using the treatment with the highest transcript levels to detect differences between the two treatments better.
Results
- In section 3.1, Figure 1: Chance A, B and C to a, b and c in the legend.
Response: Thank you very much for the comment. The figures were modified to use capital letters (A, B, C, etc..).
- In section 3.1, line 196: change ml-1 to mL-1.
Response: In the new version, this change has been made.
- In section 3.2: Plot a graph showing the measured radius of colony growth for all strains under all tested conditions. Apply statistical analysis and show if it is statistically significant.
Response: Thank you for your valuable suggestion. We acknowledge the reviewer's concern regarding the significance of our findings, particularly in cases where the differences between treatments and phenotypes are minimal and may not be convincing. However, in the context of our specific dataset and experimental results, this concern does not apply.
Our data clearly show that the ∆nik1 strains exhibit a cessation of growth at higher osmolyte concentrations, whereas the WT strain displays robust growth under these conditions. This observed contrast in growth patterns is evident to the naked eye and does not require additional demonstration for clarity.
- In section 3.2, lines 210-211: Calculate and write the percentage of the reduction compared to the parental.
Response: We appreciate the valuable suggestion provided by the reviewer. To accurately quantify the growth rate and inhibition percentage, it is necessary to measure the colony diameter before the treated and untreated (control) colonies reach the edge of the plate, typically occurring after approximately 48-60 hours of incubation. In the case of conidia, the situation is more complex due to minimal growth caused by Triton treatment and the varying number of strains on each plate.
As depicted in the figure length, this particular experiment spanned a duration of four days. Analysing the images over such an extended timeframe poses challenges, primarily due to the volume of data and the intricacies involved in precisely tracking growth dynamics.
- In section 3.2, figure 2: Identify in Figure 2a which colonies are the mutants, which mutants they are, and which colony is the wild-type.
Response: Thank you for your comment, the Figure was adjusted, including a picture indicating the strain location on the plates.
- In section 3.2, figure 2: Change A and B to a and b in the legend.
Response: Thank you for the comment. The figures were modified to use capital letters (A, B, C, etc.).
- In section 3.3: repeat the analysis of conidia growth in congo red. The different strains tested grew so much in four days that it is difficult to analyse the effects on the growth of the strains. It would be interesting to cultivate them in less time.
Response: Thank you for the reviewer's suggestion. As discussed above, the growth rate must be measured early and using fewer strains per plate. We analysed sensitivity to lytic enzymes as a quantitative assay to complement cell wall integrity analysis using Congo red.
-In section 3.3, lines 231-232: Calculate and write the percentage of the reduction compared to the parental.
Response: Thank you for the reviewer's suggestion. As discussed above, growth rate has to be measured early and with fewer strains per plate. We analysed sensitivity to lytic enzymes as a quantitative assay to complement the analysis of cell wall integrity using Congo red.
-In section 3.3, figure 3: Improve the quality of the image. Change A and B to a and b in the legend. Write in the legend what are both D and L in the image.
Response: The Figure was improved.
- In section 3.3, figure 3, line 244: the authors say "The strain order is as indicated in Fig. 2". Delete this sentence in the legend of figure 3 and write the strain order in the Figure 3.
Response: Thank you for your comment. We have updated the figure legends to reflect that the strain order aligns with the arrangement presented in Figure 2A.
-In section 3.3, figure 3, line 248: the authors say "An asterisk indicates mean values that are statistically different from the control" but in figure 3b the statistic is represented with different letters.
Response: Figure 3 in the new version has been appropriately adjusted to reflect these changes.
-In section 3.3, lines 250-252: Plot a graph showing the measured radius of colony growth for all strains under all tested conditions. Apply statistical analysis and show if it is statistically significant.
Response: We appreciate the valuable suggestion from the reviewer. As previously discussed, we recognise the importance of measuring the growth rate earlier and employing fewer strains per plate. In response to this, we have incorporated a quantitative assay assessing sensitivity to lytic enzymes, which is complementary to our analysis of cell wall integrity using Congo red. This combination of methods enhances the comprehensiveness of our study.
-In section 3.4, lines 254-255: rewrite this sentence.
Response: This sentence was rewritten.
-In section 3.4 line 279. Write that this was the negative control.
Response:In the new version, it was indicated.
-In section 3.4, figure 4: Specify what figures B and C are in the legend because these two graphs are different. In addition, statistically analyse these two graphs, showing the standard deviation of each point and showing whether the expression is statistically different at each point.
Response: We greatly appreciate your insightful suggestion. It's important to note that in Figures 4 and 5, our gene expression analysis was conducted based on data from two biological replicates, which unfortunately does not allow for robust statistical analysis. Furthermore, in our study, we adhere to a stringent criterion for categorising genes as induced or repressed in response to osmotic shock, ensuring consistency across all experiments. Specifically, we only consider genes to exhibit such behaviour if they consistently demonstrate it in every experiment.
For the chs genes in particular, the observed differences were quite subtle, necessitating a more quantitative analysis to define their regulation precisely. As a result of these considerations, we have chosen to label these findings as "suggestive cautiously."
- In section 3.4, lines 289-294: I think this statement could be phrased more concisely without repeating "Again".
Response: Thank you so much, your suggestion was considered.
- In section 3.4, line 301: Write that this was the negative control.
Response: In the new version, it has been indicated.
- In section 3.4, figure 5b: statistical analysis needs to be included, otherwise, authors cannot claim whether there is any significant difference between samples.
Response: Thank you for the suggestion. As indicated in the length of Figures 4 and 5, gene expression was analysed from two biological replicates; for that reason, we can not statistically analyse. Furthermore, we consider genes induced or repressed for an osmotic shock, only if these showed the same behaviour in all experiments. In the case of chs genes, differences detected were minimal, requiring a quantitative analysis to define better its regulation. For the before mentioned, we carefully indicate as "suggest".
- In section 3.5, lines 323-325: Can the authors revise the result narrative and discussion in this regard?
Response: Thank you for the suggestion, the narrative has been modified.
Discussion:
-In this section, the authors could also describe the roles of histidine kinase SNL1 already characterised in Trichoderma reesei.
Response: Thanks so much for your suggestion; it has been included.
-lines 344-363: Can the authors revise the result narrative and discussion in this regard?
Response: Thank you for your suggestion, it has been modified.
-lines 377-379: Can the authors revise the result narrative and discussion in this regard?
Response: Thank you for your suggestion, it has been modified.
-line 381: change chk1Δ to Δchk1.
Response: Thank you for the reviewer's suggestion, It has been changed.
- lines 397-402: Can the authors revise the result narrative and discussion in this regard?
Response: Thank you for your suggestion, it has been modified.
-line 421: perhaps it is more appropriate to change the word prove to suggest.
Response: Thank you for your suggestion, it has been modified.
Supplementary material
Figure S2 - Plot a graph showing the measured radius of colony growth for all strains under all tested conditions. Apply statistical analysis and show if it is statistically significant.
Response: We appreciate the feedback provided by the reviewer. In response to this input, we have adjusted the Figure, now clearly indicating the placement of strains in stress assays involving conidia.
However, it is essential to note that due to the extended duration of this particular experiment, it cannot be subjected to rigorous statistical analysis. Nevertheless, it's crucial to emphasise that this experiment does not impact the main conclusion regarding the relationship between osmotic stress and cell wall integrity.
Figure S2b- Identify in Figure which colonies are the mutants, which mutants they are, and which colony is the wild-type.
Response: Figure length indicates strain location in stress assays with conidia.
Figure S3 - Plot a graph showing the measured radius of colony growth for all strains. Apply statistical analysis and show if it is statistically significant. Identify in Figure which colonies are the mutants, which mutants they are, and which colony is the wild-type.
Response: We sincerely appreciate the reviewer's valuable suggestion. However, it's important to note that statistical analysis is not feasible due to the extended duration of this experiment. Nevertheless, we want to underscore that the outcome of this experiment does not impact the core conclusion regarding the relationship between osmotic stress and cell wall integrity.
We genuinely value the reviewer's input and will consider these suggestions for future research endeavours. Thank you very much for your feedback.
Reviewer 3 Report
Excelent job;
I do not have nothing to add.
Author Response
Response: Thank you so much for the reviewer's time, we appreciate a lot.